# Clinically Relevant Unsupervised Online Representation Learning of ICU Waveforms

## Abstract

Univariate high-frequency time series with real-time state changes are prominent in medical, economic and environmental applications. In the intensive care unit, for example, changes and intracranial pressure waveforms can indicate whether a patient is developing decreased blood perfusion to the brain during a stroke. However, most representation learning to resolve states is conducted in an offline, batch-dependent manner. In high frequency time-series, high intra-state and inter-sample variability makes offline, batch-dependent learning a relatively difficult task. Hence, we propose Spatial Resolved Temporal Networks (SpaRTeN), a novel composite deep learning model for online, unsupervised representation learning through a spatially constrained latent space. SpaRTeN maps waveforms to states, and learns time-dependent representations of each state. Our key contribution is that we generate clinically relevant representations of each state for intracranial pressure waveforms.

## 1 Introduction

Univariate high-frequency time series arise in several domains including economics, medicine, and environmental studies Thorsen-Meyer et al. (2020); Shakeel & Srivastava (2021); Yao et al. (2019). High-frequency time series often exhibit different states. For example, in a patient suffering from a stroke, an intracranial pressure (ICP) waveform normally flucturates somewhat. However, a transition to a state where it is persistently elevated may lead to blindness and other neurological problems Mollan et al. (2016). Early detection of this state transition may enable physicians to intervene appropriately for better outcomesLlwyd et al. (2022).

Many algorithms like shapelets, hierarchical latent factor models, hidden Markov Model-like methods, change point and anomaly detection techniques, and N-Beats are dedicated towards disentangling time series into their respective subcomponents Li et al. (2021); Grabocka et al. (2015); Oreshkin et al. (2019a); Blazquez-Garcia & Conde (2022); Aminikhanghahi & Cook (2017); Van Den Oord & Vinyals (2017) but few are dedicated towards disentangling states within a single time series Franceschi et al. (2019) or predicting future state transitions. For high-dimensional datasets, unsupervised methods like t-SNE, UMAP, and SOMs can be used to project samples into lower dimensions with spatial relationships Van der Maaten & Hinton (2008); McInnes et al. (2018). However, in time series, dimensionality is proportional to series length. As a result, state determination requires encoding time series into fixed-length vectors, followed by clustering algorithms like k-means. These methods can capture long-range dependencies but rely on non-differentiable function fitting.

Also, these methods are often offline, in that they learn from an entire training dataset at once, before being evaluated and deployed. This can be problematic, especially in the context of dataset shift or high inter-sample variability. Every time a new batch of data is received, the entire model needs to be retrained. High-frequency time series data like waveforms are often encountered in scenarios more suitable for online learning, wherein a learner attempts to tackle some predictive task by learning a sequence of data in the order they are received Hoi et al. (2021).

Extraction of states or state-transitions from a high-frequency time series requires online unsupervised representation learning, a relatively understudied field. Fuzzy neural networks create a set of modifiable rules Luo et al. (2019), but successive rule changes makes state inference relatively volatile and inconclusive. Another example of the state-of-the-art time series forecasting method is

temporal fusion transformers (TFTs), which can provide interpretable risk prediction via attention mechanisms Lim et al. (2021); Kamal et al.. This method combines feature attention with sequence attention to generate interpretable forecasts, and has shown great promise in time series forecasting.

We propose **Spa**tial **R**esolved **Te**mporal **N**etworks (SpaRTeN), a composite differentiable unsupervised deep learning network to learn a discrete spatial representation from a high frequency time series via temporal ensemble learning. We show that our method outperforms TFTs with the same number of parameters in benchmarks of online learning tasks. We also note that TFTs can be included within our method due to the flexibility of the composite model.

## 1.1 CONTRIBUTIONS

- We introduce SpaRTEn, a new framework for online learning of spatial representations from high-frequency time series.
- We demonstrate that introduction of a latent space improves rather than harms SpaRTEn's ability to forecast and cluster high frequency data in real-time, compared to state of the art models.
- We show that SpaRTeN can generate clinically meaningful representations of medical intracranial pressure waveforms.

## 2 SPATIAL RESOLVED TEMPORAL NETWORKS

### 2.1 PROBLEM FORMULATION

For the purposes of time series forecasting, we examine the problem of simultaneously learning:

1. a function $S : \boldsymbol{x_t} \rightarrow \boldsymbol{s_t}$, which maps a time series $\boldsymbol{x_t}$ of length $k$, $\{x_{t-k}, x_{t-k+1}, \ldots, x_t\}$, to a discrete state $\boldsymbol{s_t}$, and
2. a set of functions $R_{\boldsymbol{s_t}} : \boldsymbol{x_t} \rightarrow \hat{\boldsymbol{y}}_t$ which, for each state $\boldsymbol{s_t}$, map the input time series of length $k$ to a forecast time series $\hat{\boldsymbol{y}}_t = \{x_{t+1}, x_{t+2}, \ldots, x_{t+w}\}$ of length equal to the prediction window $w$.

$S$ and $R$ are optimized to maximize the probability of assigning the time series $\boldsymbol{x_t}$ to the most suitable function $R_{(.)}$ as determined by an objective function $L$:

$$\max_{S} \min_{R_{(.)}} E[L(R_{S(\boldsymbol{x_t})}(\boldsymbol{x_t}))]$$

where the expectation $E[L(.)]$ is taken over the set of all time series. This corresponds to the minmax framework described for GANs in Goodfellow et al. (2014).

### 2.2 MODEL ARCHITECTURE AND THE FORWARD PASS

We implement the above with a composite model architecture depicted in Figure 1, where $S$ and $R$ are depicted as analogously named blocks. The height $a$ and width $b$, common to the two blocks, represents the two dimensional discrete state space, which can also be considered to be the latent space for this model.

The $S$ block implements convolutional filters (Appendix A.3) to map an input time series $\boldsymbol{x_t}$ to a density over the discrete two dimensional space of states (green arrow 1). The $R$ block consists of a spatially arranged ensemble of LSTM sub-networks, each of which makes a forecast for the input $\boldsymbol{x_t}$ with a prediction window of $w$ (blue arrow 1). For each input $\boldsymbol{x_t}$, the sub-network in $R$ corresponding to the greatest density output by $S$ (green arrow 2) is used for generating the prediction $\hat{\boldsymbol{y}}_t$ (blue arrows 2, 3).

We choose to place $\boldsymbol{s_t}$ in a two-dimensional discrete state-space $(i, j)$, because it facilitates easy visualization of time series corresponding to individual states, which previous methods like SOM-VAE and TFT are unable to currently do. We can parameterize the number of states by adjusting the width and height of the latent space, $a$ and $b$. The state space width and height are hyper-parameters that should be adjusted depending on the *a priori* assumptions of dataset complexity.

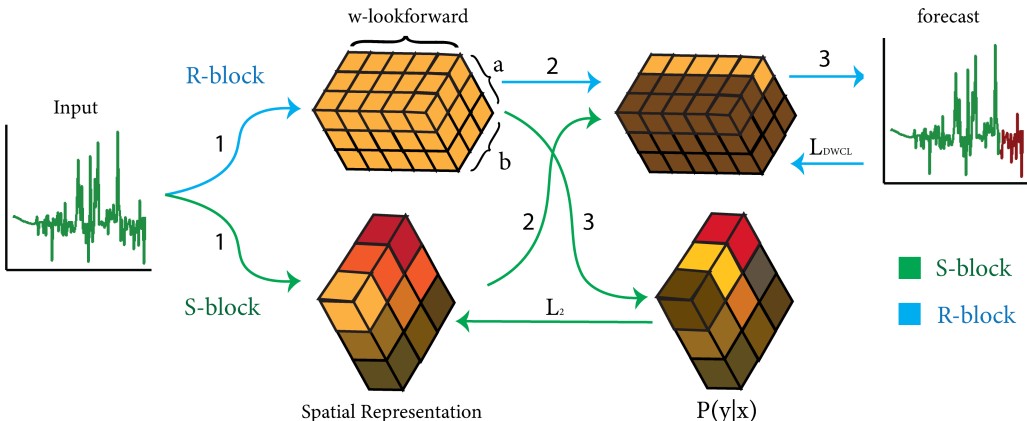

Figure 1: **Schematic representation of model architecture.** Blue is forward propagation and green is back-propagation. Time series from the data space are forecast by the sub-networks in an $R$-block with a look-forward period of $k$. The dimensionality of the spatial representation is a $a \times b$. Simultaneously, an S-Block predicts the most relevant sub-network for the time series. The predicted sub-network is used to generate a forecast and is then back-propagated via Equation **??**.

### 2.3 LOSS FUNCTIONS AND TRAINING WITH BACKPROPAGATION

When implemented as described here, the entire system can be trained with backpropagation. Two distinct loss functions have to be accounted for, for the $S$ and $R$ blocks.

Sub-networks of $R$ can have a primary objective ($L_{\text{objective}}$) of forecasting, classification or re-construction if we assign $\boldsymbol{y_t}$ to be the forecasting window, a labeled class or $\boldsymbol{x_t}$, respectively. For the examples in this paper, we consider the objective to be forecasting. To further impose structure, we introduce a second objective inspired by contrastive loss and self-organizing maps, the distance-weighted contrastive loss ($L_{DWCL}$) (Appendix A.1). For a single sample, we define the loss function for to be

$$L_{R_{\boldsymbol{s_t}}} = L_{\text{objective}}(\boldsymbol{y_t}, R_{\boldsymbol{s_t}}(\boldsymbol{x_t}; \theta_{R_{\boldsymbol{s_t}}})) + \alpha \times L_{\text{DWCL}}(R_{\boldsymbol{s_t}}(\boldsymbol{x_t}; \theta_{R_{\boldsymbol{s_t}}}), R(\boldsymbol{x_t}; \theta_R)) \quad (1)$$

$$L_{\text{DWCL}}(R_{\boldsymbol{s_t}}, R) = -\log \frac{e^{R_{\boldsymbol{s_t}}}}{E_{\boldsymbol{z} \sim Z}\left[e^{\text{sim}(R_{\boldsymbol{z}}, R_{\boldsymbol{s_t}})} \times ||\boldsymbol{s_t} - \boldsymbol{z}||_2\right]} \quad (2)$$

where $\boldsymbol{y_t} = (x_{t+1}, x_{t+2}, ..., x_{t+w})$, the ground truth values from the time series, $\alpha$ is a learned hyper-parameter to modulate the relative effects of the two terms, $\boldsymbol{z}$ is a state drawn from the set of all states $Z$, sim is a metric of similarity between $R_{\boldsymbol{z}}$ and $R_{\boldsymbol{s_t}}$, such as a normalized dot product or cosine-similarity.

$L_{DWCL}$ forces similar states to be closer and dissimilar states to be separated, causing the waveforms to cluster in the state space. During each forward propagation step, the predictions are generated by each forecasting network in the R-block. After the predictions are generated by each network in the $R$ block, the distance-weighted contrastive loss is created by calculating the difference in the predictions between each of the network in the R-block and network selected by the $S$ block, and weighting it by Euclidean distance of the network and the selected sub-network (blue leftwards arrow).

The $S$ block has a separate loss function $L_s$, contingent on its objective, which is to predict the spatial state occupied by the next time step. It can be conceptualized as the distance between $S$ block prediction of the best state and the network in the $R$ block with the lowest error with respect to $\boldsymbol{y_t}$ (green leftwards arrow):

$$L_s = \|S(\boldsymbol{x_t}) - \arg\min_{\boldsymbol{s_t}} L(R_{\boldsymbol{s_t}}(\boldsymbol{x_t}))\|_2 \quad (3)$$

While an $L_1$ or $L_2$ norm may better capture the information about the density of the spatial networks, in practice it may not provide a sufficient gradient for $S$ to learn well (Appendix A.3). Early

in learning, when the sub-networks in the R-matrix perform poorly, $S$ displays highly unstable dynamics, which in turn hinders $R$-block learning. Rather than training $S$ to directly maximize the spatial density, we can improve stability by treating the objective as a classification problem and minimize the negative log likelihood (details in Appendix A.2).

The full training algorithm is provided in Algorithm 1.

## 2.4 Ensemble Weight Sharing

Inductive transfer learning leverages an inductive bias to improve performance on a target task and eliminates redundant learning of patterns in data structure Zhuang et al. (2021). To generate sub-networks with weights that represent distinct states rather than shared structure between states in the time series, we employ an inductive transfer learning framework. This procedure increases the gap between the sub-network posteriors, which further enhances the contrastive learning aspect of the network. From an information theoretical perspective, the process of learning a shared posterior can be thought of as a lossless compression of the hidden states by encoding them into a shared embedding. In turn, non-unique learning of state-independent behavior only needs to take place once rather than $a \times b$ times.

While mode collapse is a known problem, we find that the sharing of weights across the first few layers leads to robust performance as seen with the paradigm of transfer learning. This is quite unlike the mode collapse seen in the training of generative adversarial networks. Without the sharing of weights across the first few layers, we find that learning requires significantly more samples because the overall structure of the time series must be learned for each unit in the $R$ block, in addition to learning of the relevant state. In contrast, with weight sharing, we find that learning of the overall structure of time series can be done jointly, and the state separation can be learned by each sub-network.

In our implementation of the SpaRTeN framework, this auxiliary network maps the hidden states of $R$, which is an $h \times a \times b$ embedding to a low-dimensional embedding $h'$, which is subsequently appended to a dense layer of each sub-network. This procedure ensures that the shared weights are differentiable during training. After back-propagation, weights are copied to all sub-networks.

---

**Algorithm 1** Spatial Projection of Time Series with Temporal Ensembles

---

**Require:** $\{x_0, \ldots, x_T\}$, where $T$ is the length of the time series. $\boldsymbol{x_t}$ represents $\{x_{t-k}, \ldots, x_t\}$ where $k$ is the look-back period, and $y_t$ represents the forecast $\{x_{t+1}, \ldots, x_{t+w}\}$ where $w$ is the forecast window. Assign a width and height to the state space $a, b \in \mathbb{Z}^+$. Randomly initialize weights of the $R$ and $S$ block: $\theta_R, \theta_S \sim N(0, 1)$.

  **for** $m = k$ to $m = T - w$ **do**
    $\boldsymbol{x_m} \leftarrow \{x_{m-k}, \ldots, x_m\}$
    $\boldsymbol{y_m} \leftarrow \{x_{m+1}, \ldots, x_{m+w}\}$
    $(\hat{i, j}) = S(\boldsymbol{x_m}; \theta_S)$
    $\hat{\boldsymbol{y}}_{\boldsymbol{m}(i,j)} = R_{(i,j)}(\boldsymbol{x_m}; \theta_R) \forall \{i : [0, a), j : [0, b)\}$
    Update $R_{(\hat{i,j})}$ via gradient descent on $L_{DWCL}(\hat{\boldsymbol{y}}_{\boldsymbol{m}(\hat{i,j})}, \boldsymbol{y_m})$
    Update $S$ via gradient descent on $L(S(\boldsymbol{x_m}), \arg\min_{i,j}(L(\hat{\boldsymbol{y}}_{\boldsymbol{m}(i,j)}, \boldsymbol{y_m}))$
  **end for**

---

## 2.5 Related Work

### 2.5.1 Twin Neural Networks and Boostrap Your Own Latent Space

Twin neural networks and Bootstrap Your own Latent Space (BYOL) contain two or more sub-networks He et al. (2018); Grill et al. (2020), and can learn semantic similarity between different samples. Subnetworks cast as recurrent neural networks have been used to learn and visualize time series similarities Pei et al. (2016). Like twin neural networks and BYOL, our framework employs contrastive loss with subnetworks. However, there are a few key differences.

First, SpaRTeN learns relationships between states in an online, sequential manner rather than offline and batch-dependent. As a result, SpaRTEn generates states that are time-dependent, whereas

implementations of BYOL are time-independent. When the learned network is time-independent, we may see a different state assigned to each sequential time point or punctuated changes in state rather than a gradual evolution Fortuin et al. (2018). However, clinically relevant changes in state are usually time-dependent, and as such, SpaRTen is expected to generate more interpretable predictions. For example, a patient who is developing an infection deteriorates over time until reaching a state of sepsis, rather than abruptly changing from a healthy to a septic state.

Second, online learning captures relationships such as stationary versus non-stationary time series that are not easily ascertained in time-agnostic state encodings. For example, in a stationary time series, we notice that after a state has stabilized, the loss function for network belonging to that state slowly decreases. In non-stationary time series, patterns associated with the state start diverging from the expected pattern and the network loss gradually increases until it reaches a threshold. At the threshold, a state transition occurs and a new network is selected to represent that given state, implicitly allowing the detection of the state change. These relationships cannot be deciphered in time-step agnostic state-encodings.

### 2.5.2 TEMPORAL ENSEMBLES AND MIXTURE OF EXPERTS

Ensemble Learning refers to a family of techniques where multiple learners are trained to solve the same problem Zhou (2009). Ensemble methods construct multiple hypotheses from these base learner algorithms and join them to generate a prediction that generalizes much better than the individual algorithms. Ensembles with base learner LSTMs have been used on financial time series forecasting to improve performance Sun et al. (2018). Our framework forces base learners to occupy a Euclidean space, which can subsequently be used to generate interpretable representations. Other online unsupervised methods with time series have developed composite or adaptive model approaches focused on anomaly detection followed by model adaptation Karaahmetoglu et al. (2020); Savitha et al. (2020). Having distinct sub-networks for each state allows for different models to uniquely represent distinct states.

The key advantage of utilizing the framework proposed in the paper over a mixture of experts is the idea of state separation, which allows visualization and explainability via representation. Utilizing the novel contrastive function to promote diversity in recurrent neural networks allows for state separation. This is particularly relevant in the medical setting, where state separation can allow for the identification of clinical states amenable to different diagnostic and therapeutic interventions. Learning these representations can allow a physician to give a drug or an economist to formulate a fiscal policy. We provide the specific example of the ICU measurements, where if an individual belongs to a state where increased pressure on the brain is identified, then an appropriately targeted intervention can be provided. Mixtures of expert models do not typically generate representations to interpret and therefore, have limited explainability in state-dependent time-series analysis.

### 2.5.3 VAEs AND TFTs FOR DISCRETE LATENT SPACES

Deep Learning Networks with discrete latent spaces have been utilized in the past with relatively high degrees of success. For example, VAEs can discretize the latent space with an encoder-decoder setup, and has been more heavily applied to interpreted disentangling of discrete representation learning Williams et al. (2021). Networks with attention such as temporal fusion transformers encode a time series patterns into a key-value dictionary, which can be thought of as another discrete latent space Lim et al. (2021).

SpaRTeN offers advantages over these methods. First, while VAEs and TFT generate a representation in the latent space, SpaRTEn clearly identifies the representation in the space of the time series. Generating a representation in the same space as the time series allows for improved explainability, as we demonstrate in section 3.4. Second, VAEs are typically constrained to reconstruction or KL-divergence based loss and have yet to be implemented for forecasting. In contrast, SpaRTeN can be adapted to different tasks by choosing different loss functions for the $R$ block. Third, our benchmarks as a versatile ensemble, SpaRTEn can include these models as networks in the R-Block. Fourth and finally, we benchmark against TFTs forecasting ability and show that SpaRTEn outperforms.

One extension of VAEs with a spatially resolved latent space encodes time series in a self-organizing map Fortuin et al. (2018). Self-organizing maps are an extension of discrete latent spaces that

represents an input space with fixed dimensionality as a discrete two-dimensional Euclidean space. Each node in the two dimensional map is a single neuron, and the best matching neuron is adjusted towards input. This model learns state transitions via Markov modeling on the self-organizing map. We extend self-organizing maps differently, where nodes represent distinct subnetworks rather than a decodable state, which allows distinct weights and architectures. Using a separate block to predict the node, we can eliminate the Markov chain used in SOM-VAEs.

## 3 APPLICATION

For our applications, we focus on three distinct tasks involving high frequency time series — online forecasting, zero-shot clustering and clinically significant representation learning.

### 3.1 ONLINE FORECASTING WITH SPARTEN

Online forecasting involves looking at a window in a time series and using that window to predict the next set of elements. For the R-Block, we minimize sMAPE (Symmetric Mean Absolute Percentage Error) as the primary objective in a forecasting task:

$$\text{sMAPE} = \frac{1}{N} \sum_{i=1}^{N} 2 \times \frac{|\boldsymbol{y_i} - \hat{\boldsymbol{y_i}}|}{|\boldsymbol{y_i}| + |\hat{\boldsymbol{y_i}}|} \qquad (4)$$

where $N$ is the number of examples used for training. sMAPE is a metric that has been typically reported in the past with competitions like the M4 time series forecasting competition Makridakis et al. (2018). For the S-Block, we utilize a standard cross-entropy loss for multi-class classification.

We benchmark against SOTA online forecasting models with convolutional approaches such as N-Beats Oreshkin et al. (2019b) and attention based methods like Temporal Fusion Transformers Lim et al. (2019) and Autoformers Wu et al. (2021), where SpaRTEn outperforms on three out of the four datasets (Table 1). Because these forecasting remains relatively unstable over these datasets, we only include RMSE calculated during which the S-block predictions were stable. Over large batches, we anticipate that other networks such as autoformers and TFTs that can capture long-term dependencies and recurrent patterns would outperform. However, these networks are not mutually exclusive with our ensemble method, and we utilize a base case of simple auto-regressive networks to demonstrate that spatially constrained ensemble learning does not hurt performance.

Table 1: RMSE of benchmarks on an online forecasting task

| Model | Datasets | | | |
|---|---|---|---|---|
| | Electricity | Traffic | Stocks | Retail |
| LSTM | 2.93 | 32.10 | 0.13 | 13.76 |
| N-Beats | 2.84 | 3.10 | **0.10** | 14.16 |
| TFT | 2.49 | 15.10 | 0.11 | 13.87 |
| Autoformer | 6.61 | 3.34 | 1.24 | 4.98 |
| SpaRTEn | **1.57** | **1.58** | 0.67 | **1.59** |

### 3.2 ZERO-SHOT CLUSTERING OF HIGH FREQUENCY TIME SERIES

Zero-shot clustering is an unsupervised method that involves classifying the time series the first time it is seen without information corresponding to a label. We demonstrate that SpaRTeN can generate prototypical waveforms, that can be utilized by K-Nearest Neighbors to perform state-of-the-art for zero-shot clustering methods. We benchmark on traditional clustering techniques such as KNN, Gaussian Mixture Modeling and Spectral Clustering.

In order to compare the distinct properties of each state and demonstrate the ability to cluster waveforms within a given state, we train SpaRTeN with a latent state space of $3 \times 3$. We train a KNN with $k = 9$ on the 9 ($3 \times 3 = 9$) distinct waveforms aggregated by state, and evaluate its ability to cluster all the waveforms on the dataset using a silhouette score, which is widely used to evaluate

the goodness of a clustering technique. SpaRTEn outperforms all other methods used to cluster time series on all four datasets.

Table 2: Silhouette score of benchmarks on an unsupervised clustering task

|  | Datasets | | | |
| --- | --- | --- | --- | --- |
| Model | Electricity | Traffic | Stocks | Retail |
| Random | 0.023 | 0.22 | 0.012 | 0.011 |
| Spectral | 0.005 | 0.09 | 0.005 | 0.002 |
| GMM | 0.024 | 0.12 | 0.011 | 0.010 |
| SpaRTEn | **0.028** | **0.24** | **0.026** | **0.027** |

## 3.3 ABLATION STUDIES

We run additional ablation experiments across these four datasets including increasing the size of the latent space, and eliminating the distance-weighted contrastive loss, which lead to deterioration on the clustering metric.

These ablation studies demonstrate that the S-block is necessary for adequate performance in online forecasting, which may be due to over-squashing in naive temporal models Alon & Yahav (2020). Other aspects of model are less important for this metric.

In contrast, all aspects of the model are crucial to generating clusters. For example, eliminating the distance-weighted contrastive loss leads to a 15% decrease in silhouette score for UCI Electricity, a 29% decrease for UCI Traffic, and a 53% decrease for Kaggle Retail. Although the online forecasting aspect of SpaRTen performed poorly in the Oxford stocks dataset, the clustering task beat the benchmarks. Eliminating the distance-weighted contrastive loss led 51% decrease in silhouette score on this task. We hypothesize that adding a spatial loss improves sub-network diversity by increasing state separation, aiding downstream clustering tasks. We also find that an over-parameterized the state-space can actually lead to self-regularization, where the network learns to under-utilize the coordinate system given the complexity of the time series.

Table 3: Ablation study on benchmarked datasets

|  | Dataset | | | | | | | |
| --- | --- | --- | --- | --- | --- | --- | --- | --- |
|  | Electricity | | Traffic | | Stocks | | Retail | |
| **Ablation** | RMSE | Silhouette | RMSE | Silhouette | RMSE | Silhouette | RMSE | Silhouette |
| S-block | 2.93 | **X** | 32.10 | **X** | **0.13** | **X** | 13.76 | **X** |
| 10 × 10 | 2.58 | 0.019 | **1.58** | 0.15 | 0.57 | 0.005 | 1.61 | 0.001 |
| DWCL | 2.57 | 0.024 | 1.62 | 0.17 | 0.60 | 0.012 | 1.60 | 0.013 |
| None | **1.58** | **0.028** | **1.58** | **0.24** | 0.67 | **0.026** | **1.59** | **0.027** |

## 3.4 TIME IS BRAIN: CLINICALLY RELEVANT REPRESENTATION LEARNING

We demonstrate the practical application of SpaRTeN to intracranial pressure (ICP) waveforms from the MIMIC-IV dataset. ICP waveforms are suited for online learning because they provide high frequency time series with a significant inter-individual variability. In neuro-critical care, time is brain Desai et al. (2019). Timely identification of ICP states may allow treating physicians to intervene and mitigate neurological injury.

We trained SpaRTeN on ICP waveforms across 400 time steps (Figure 2), optimizing hyper-parameters with a grid-search conducted over the state-space and learning rates for both the $R$ and $S$ blocks, $\alpha$, and the depth and width of the sub-networks. This resulted in 9 distinct classes (Figure 2(c)). SpaRTeN outperformed spectral clustering, k-means applied random sampling and Gaussian mixture models as determined by the silhouette score, and this was robust to sample size (Figure 2(b), Table 4).

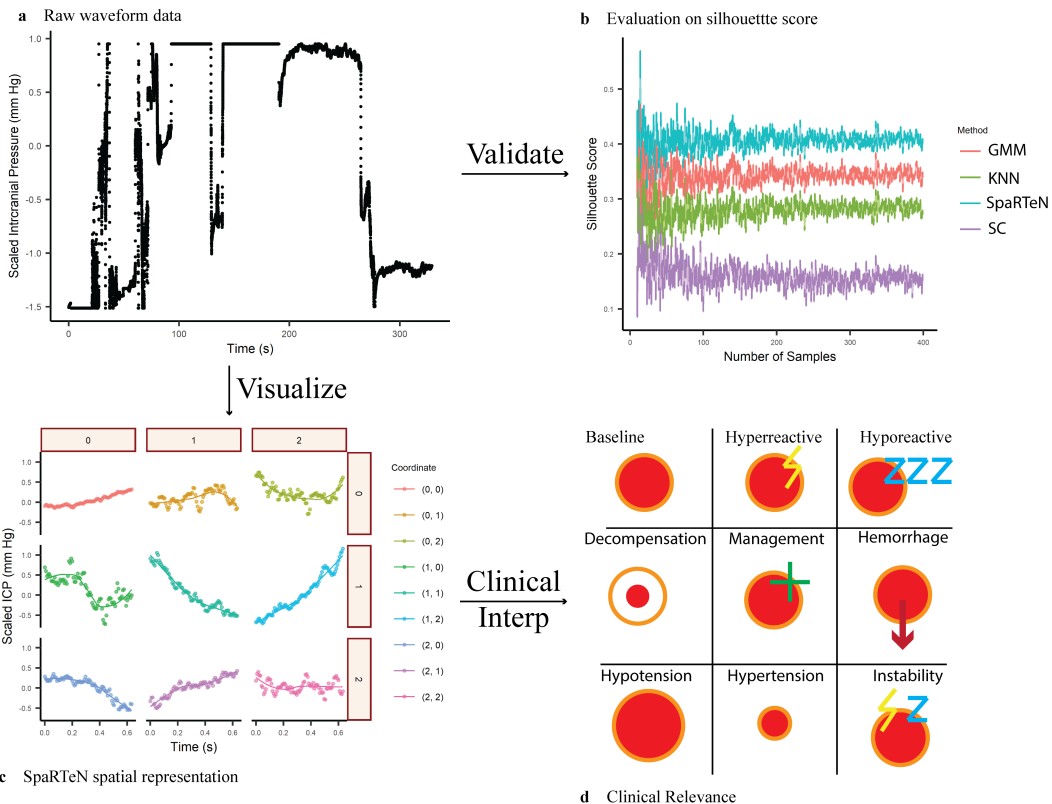

Figure 2: **Qualitatively and quantitatively evaluating model representations.** a) Raw waveforms of Intra-cranial Pressure with high intra-sample variability. b) Bootstrapped (10x) results of silhouette score across different sized samples demonstrates that SpaRTeN outperforms other clustering methods. 95% Confidence intervals reported in the figure, but may be too small to see. c) Clusters generated by SpaRTeN represent distinct trends within the time series. d) Clinical interpretation of each of the waveforms.

The nine classes identified by SpaRTeN on are interpretable as clinical states (Figure 4(d)).

At position $(0, 0)$, we see that the waveform is both stable and relatively constant. This is a strong baseline for what non-pathological waveforms should look like. At point $(0, 1)$ we see increasing oscillations in the pressure, but with a smaller average change. This can be part of an adaptive response to injurySvedung Wettervik et al. (2020), where some of the compensatory mechanisms are starting to be hyper-reactive. In contrast, at point $(0, 2)$, we see that the intracranial pressure waveform has a U-shape, with a relatively slow dip and return to baseline. Such slow, hypo-reactive ICP waveforms have been associated with worse performance on the Glasgow Coma Scale Tian et al. (2013), a measure of level of consciousness. A physician may try to shift a patient's state from $(0, 2)$ to $(0, 1)$ in order to improve outcomes by increasing the tone of the sympathetic nervous system Schmidt et al. (2018).

In the second row represents acute dysregulation of ICP. At point $(1, 1)$, we see that the there are signs of instability, followed by an acute drop in the intracranial pressure. Such a patient may benefit from an urgent CT scan. At point $(1, 1)$ and $(1, 2)$, we see that there is complete dysregulation of the brain's vasculature with dramatic decreases and increases in intracranial pressure, respectively. $(1, 2)$ may represent cerebral hemorrhage which causes a rapid increase in ICP. Physicians can manage this with various interventions or surgeryCaceres & Goldstein (2012). $(1, 1)$ shows a rapid decrease in intracranial pressure as might be expected following the placement of an external ventricular drainKramer (2021).

The bottom row represents slower, more chronic decompensation than seen in the second row. $(2, 0)$ and $(2, 1)$ show slower decreases and increases in ICP, respectively. While both these can manifest as headaches, diuretic drugs can worsen the former but effectively treat the latter Luetzen et al. (2021). Finally, the pattern in $(2, 2)$ shows slower oscillations without much average change but may be an early warning of progression to one of the more acute states in the second row Oernbo et al. (2022).

Table 4: Silhouette Score of representations of Intracranial Pressure Waveforms

| Model | Sample Size | | | |
| --- | --- | --- | --- | --- |
| | 10 | 25 | 100 | 400 |
| Spectral | $0.131 \pm 0.023$ | $0.165 \pm 0.008$ | $0.109 \pm 0.007$ | $0.156 \pm 0.003$ |
| Random | $0.366 \pm 0.018$ | $0.275 \pm 0.007$ | $0.252 \pm 0.005$ | $0.276 \pm 0.002$ |
| GMM | $0.341 \pm 0.019$ | $0.345 \pm 0.005$ | $0.329 \pm 0.004$ | $0.344 \pm 0.003$ |
| **SpaRTeN** | $\mathbf{0.415} \pm 0.020$ | $\mathbf{0.422} \pm 0.006$ | $\mathbf{0.385} \pm 0.005$ | $\mathbf{0.405} \pm 0.002$ |

To our knowledge, this is the first algorithm to use an online contrastive learning approach for time series classification. While it is known that discriminative region-based zero-shot learning in images can preserve context information Narayan et al. (2021), SpaRTeN representations quantitatively capture variations within the data, and qualitatively provide key clinical insights into waveform patterns.

## 4 POTENTIAL LIMITATIONS

SpaRTeN is a novel min-max framework for decoding states, and has many of the same advantages and disadvantages as other min-max frameworks. Without sufficient gradient-based optimizations like smoothing and replacing density-based losses with negative log-likelihood losses, the gradients and states learned by SpaRTeN can be highly unstable (Appendix A.2). Subsequently, a collapse in the gradients on one of the blocks can be highly detrimental to other blocks.

Second, many datasets, especially in the ICU contain multi-modal sources of information. Currently, models like temporal fusion transforms can better account for multi-modal trends in time series and combine categorical with continuous variables. We anticipate further development of the SpaRTeN framework by including R-blocks that are capable of accounting for different variable types and data modalities may further enhance the ability of SpaRTeN to generate multi-modal archetype waveforms, which can be subsequently used to qualitatively evaluate changing states in the clinical setting.

Third, we selected 2D geometry because it was computationally tractable in terms of the distance-weighted contrastive loss, and interpretable in the ICU setting. Ablation of the the distance-weighted contrastive loss leads to poorer representation learning and clustering. Future work could explore higher-dimensional latent spaces and hyperbolic geometry.

## 5 CONCLUSIONS

We introduce a novel method called SpaRTeN (Spatially Resolved Temporal Networks) for discrete representations of time series via unsupervised learning and a forecasting objective. SpaRTeN stores models rather than samples in an embedding space, which allows for rapid interpretable learning of structured representations in high frequency time series. We show that it can be broadly applied to online forecasting and clustering. We anticipate that improvements in size, algorithmic and optimization details will only continue to further improve upon the SpaRTeN framework. Finally, we further intend to demonstrate the applicability of this model architecture to real-time, online clinical decision support in situations like the decoding states for patients in the ICU. Thus, the SpaRTeN framework can be generalized to different network blocks, optimization techniques and use cases. It takes a key step towards the goal of generating individualized state representations with online learning. These analyses demonstrate that SpaRTEn is able to decipher clinically meaningful states. Moreover, utilizing these state analyses to better disentangle states can improve the understanding of clinical treatment and associated outcomes Samartsidis et al. (2018).

## 6 REPRODUCIBILITY STATEMENT

All experiments were performed with PyTorch. The code for the algorithm is attached in the supplementary material.

## 7 ETHICS STATEMENT

Experiments with publicly available de-identified data from the MIMIC-III Waveform Database were conducted with IRB approval. All other datasets that we used are also publicly available. This work is not expected to lead to negative societal implications.

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

# A  APPENDIX

## A.1  DISTANCE-WEIGHTED CONTRASTIVE LOSS

The general idea of contrastive loss is to preserve neighborhood relationships between data points by minimizing the distance between similar points and maximizing the distance between points of different classes Hadsell et al. (2006). The general form of the contrastive loss function is the following:

$$L_{\text{contrastive}}(x_i, x_j; \theta) = \frac{1[y_i = y_j]}{2} d(f_\theta(x_i), f_\theta(_j)) + \frac{1[y_i \neq y_j]}{2} \max(0, \epsilon - d(f_\theta(x_i), f_\theta(x_j)) \quad (5)$$

where $x_i$ and $x_j$ are two distinct samples, $f$ is a function that maps $x \to R^k$, an embedding of dimensionality $k$, $d$ is a distance metric, and $\epsilon$ is the distance to the margin. In multi-class classification problems, this can be further extended as a classification problem with $K + 1$ categories He et al. (2019).

$$L_q = -\log \frac{e^{\text{sim}(f_\theta(x_t), f_\theta(x_j))/\tau}}{\sum_{k=1}^{N} e^{\text{sim}(f_\theta(x_k), f_\theta(x_j))/\tau}} \quad (6)$$

where $\text{sim}(f(x_i), f(x_j))$ is a metric of similarity between $f(x_i)$ and $f(x_j)$ and $\tau$ is the normalization factor.

We can extend this to forecasting where the positive example can be thought of as the selected sub-network, whereas the negative examples are the irrelevant sub-networks. Finally, we add a normalized distance metric, to ensure sub-networks that are closer in euclidean space have closer representations.

$$L_{\text{DWCL}}(R_{i,j}, R) = -\log \frac{e^{R_{i,j}}}{E_{x,y \sim \mathbb{Z}^{2+}}\left[e^{\text{sim}(R_{x,y}, R_{i,j})} \times \frac{\sqrt{(x-i)^2 + (y-j)^2}}{\sum_{x,y}^{\mathbb{Z}^{2+}} (x-i)^2 + (y-j)^2}\right]}$$

We visualize this further in Figure 3.

The overall computational cost is $O(c(f, b) + (K - 1) \times c(f))$, where $c(f)$ is the cost of forward propagating, and $c(f, b)$ is the cost of forward and back-propagating, and $K$ is the total number of blocks. Thus, computational cost scales with the number of sub-networks.

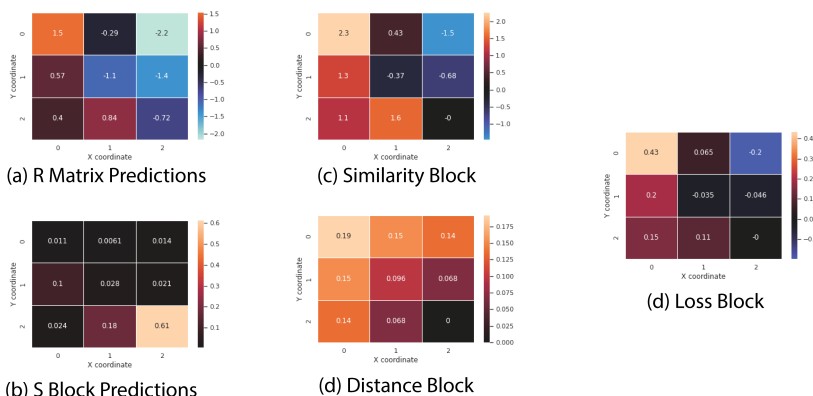

(a) R Matrix Predictions

(c) Similarity Block

(d) Loss Block

(b) S Block Predictions

(d) Distance Block

Figure 3: Loss calculations for the R-matrix. (a) Is the calculation of the R-matrix predictions for the next time step. (b) S-Block predicts the appropriate state for the next time step. (c) The similarity block calculates similarity between the chosen state prediction and the other states. (d) The distance block calculates the distance between each other state and the selected substates. The loss block is the dot product of the distance and similarity block. The loss block is summed to produce the final loss value

A.2    ENCOURAGING SMOOTHNESS OVER TIME

The goal is to predict the development of a time series in an interpretable way. This means that we may have a tradeoff between stable network dynamics and representation of a ground truth density. Learning a probabilistic model in a high-dimensional continuous space can be challenging, which necessitates the use of reductionist frameworks to improve interpretability.

Previous work in Markov chain modeling penalized state transitions via an additional smoothness term Fortuin et al. (2018). Other methods have focused on incorporating quantile outputs to maximize the signal-to-noise ratio Lim et al. (2021).

We find that by converting an $L_2$-norm-based loss function to cross-entropy loss, we can improve the stability of both the $S$-block representations, and by extension, the $R$-block ensemble:

$$L_S = - \sum_{i,j}^{\mathbb{Z}^{2+}:[a,b]} \arg\min_{i,j} (R_{i,j}(\boldsymbol{x_t}) - x_{t+1})^2 \times \log \sigma(S(\boldsymbol{x_t})) \tag{7}$$

where $\mathbb{Z}^{2+}$ is a discrete two-dimensional space of integers in $[a, b]$, the sum is over all the coordinates in the space, $R_{i,j}(\boldsymbol{x_t})$ is the prediction of the next time step by the network based on the previous time step, $x_{t+1}$ is the next step. $\sigma$ represents soft-max function, and $S(\boldsymbol{x_t})$ is the predicted state of the next time step. If S fails to provide strong initial gradients, as in the case with $L_2$-norm, then the instability of the network prevents a single sub-network from learning the characteristics of a given state (Figure 4). In turn, this causes the S-block to be increasingly volatile, which can in turn further destabilize the R-block.

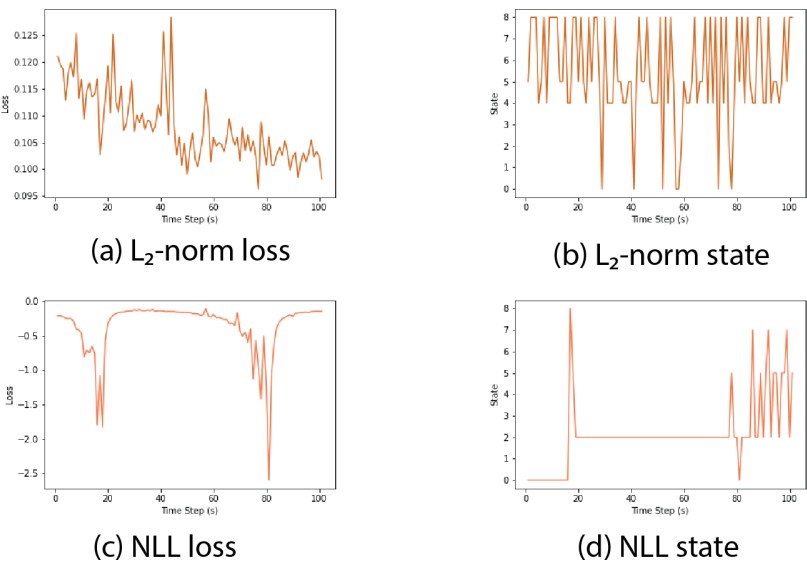

(a) L₂-norm loss

(b) L₂-norm state

(c) NLL loss

(d) NLL state

Figure 4:  Losses and associated states for $L_2$-norm and negative log-likelihood. (a) $L_2$-norm loss is significantly smaller and signal-to-noise ratio is smaller than (c) negative log likelihood (NLL) loss. The corresponding states calculated by the (b) $L_2$ loss are far more unstable than the states calculated by the (d) negative log likelihood S-block.

### A.3    S-BLOCK ARCHITECTURE

The goal of the $S$-block is to translate a high-frequency time series into a spatial coordinate system with a dimensionality of $a, b$. The flexibility of fully connected networks in conjunction with spatial constraints imposed by convolutional filters biases the network towards a spatial representation of the temporal networks.

The S-block consists of four key layers, a 1D-CNN, a fully connected network layer with $(a + 2) \times (b + 2)$ number of units, a layer that reshapes the fully connected network block into an $(a + 2) \times (b + 2)$ rectangle, followed by a $3 \times 3$ convolution with a stride length of 2, to produce an ultimate output layer of dimension $ab$ (Figure 5).

A discrete state space was chosen to improve the interpretability of the model sub-networks to produce meaningful results. However, future work may replace the discrete state space output with a representation of a density distribution or a continuous vector space.

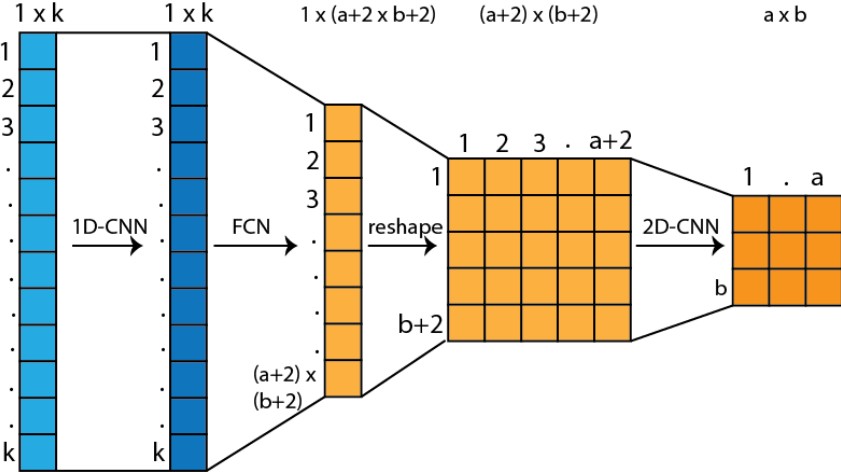

Figure 5: Network architecture of the S-Block. The S-block consists of four key layers, a 1D-CNN, a fully connected network layer with $(a + 2) \times (b + 2)$ number of units, a layer that reshapes the fully connected network block into an $(a + 2) \times (b + 2)$ rectangle, followed by a $3 \times 3$ convolution with a stride length of 2, to produce an ultimate output layer of dimension width length.

