# OpenReview forum: "Spatially Resolved Temporal Networks: Online Unsupervised Representation Learning of High Frequency Time Series"
_ICLR.cc/2023/Conference — Submitted to ICLR 2023_

### Official Review · Reviewer_AkMC · 2022-10-23

**Confidence:** 3
**Correctness:** 3
**Technical Novelty And Significance:** 3
**Empirical Novelty And Significance:** 2
**Recommendation:** 5

**Clarity, Quality, Novelty And Reproducibility:**

Clarity needs improvement to make the whole framework easy to follow and understand.
Novelty looks pretty good. However, I only see empirical support and there are no theoretical guarantees.
Reproducibility is feasible with the submitted code.


**Strength And Weaknesses:**

Strength
The whole framework looks novel and incorporates recent advances in ML like contrastive learning for time series forecasting.

Weakness
Motivations and rationale behind the proposed model is not well explained. It is not very clear how these components interact and why all the components work in a synergetic way.

**Summary Of The Paper:**

In this paper, the authors propose a new time series forecasting model for univariate high-frequency time series. There two major components/blocks of the model: one block contains ensemble of base predictors/LSTM for time series forecasting and the second spatial block is trained using contrastive learning to find the best base predictor or LSTM. The whole model works like the minimax framework. Experiments on online forecasting and zero-shot clustering show efficacy of the proposed method.


**Summary Of The Review:**

1. Regarding contrastive loss, we know that it works fine for images. Here the motivation and rationale of employing the distance-weighted contrastive loss DWCL for univariate time series forecasting is not clear to me. How are similar and dissimilar pairs selected in the training process?
2. What is the benefit of using a two-dimensional discrete state-space in the S-block? For the purpose of selecting the best base predictor, one-dimensional state-space should be enough. Also in the training of the S-block, cross-entropy loss function is used which actually works fine with the one-dimensional state-space.
3. In Sec. 3.3, why does ensemble weight sharing help for time series forecasting? If we wanna a diverse ensemble which can cover a broader state-space, should we limit the use of ensemble weight sharing which may lead to the mode collapse (common challenge in GAN).
4. The S-block actually reminds me the attention mechanism which can also be used to select the optimal state/predictor. It would be interesting to see discussion on comparison of S-block and attention. For example, what are the advantages of S-block over attention and gating in the temporal fusion transformer (TFT).
5. In practice, individual LSTM has limited forecasting power compared with state-of-the-art (SOTA) algorithms for time series forecasting. More analysis on how ensemble of LSTMs could achieve better prediction performance will help reader to understand the contribution and novelty of this work.
6. Evaluations are limited and only done on two datasets which makes it less convincing. More experiments and comparisons with SOTA methods are encouraged

---

> ### Author Response · Authors · 2022-11-19
> **Author response (1/3)**
>
> We thank the reviewer for their constructive comments, and we hope we have provided sufficient responses.
>
> 1. Regarding contrastive loss, we know that it works fine for images. Here the motivation and rationale of employing the distance-weighted contrastive loss DWCL for univariate time series forecasting is not clear to me. How are similar and dissimilar pairs selected in the training process?
>
> **Distance-weighted contrastive Loss:** We have added an explanation. The idea is that distance-weighted contrastive loss is a self-supervised loss where networks that are ‘spatially’ closer together should have more similar predictions whereas networks that are further apart should have more different predictions. The motivation and rationale for this distance-weighted contrastive loss is two-fold: first, it generates interpretable predictions, and second, it improves cluster separation as measured by the silhouette score. We have added the following excerpts to illustrate the effect of this contrastive loss.
>
> **Explanation of the distance-weighted contrastive loss:** (Section 2.3) This loss pushes sub-networks to have distinct predictions. The distance-weighted contrastive loss for univariate time series learns similar and dissimilar pairs in a self-supervised manner. During each forward propagation step, the predictions are generated by each forecasting network in the R-block. After the predictions are generated by each network in the R-Block, the distance-weighted contrastive loss is created by calculating the difference in the predictions between each of the network in the R-block and network selected by the S-Block, and weighting it by Euclidean distance of the network and the selected sub-network. Similar pairs can be thought of networks that are closer to the selected network in Euclidean space, and dissimilar pairs are those that are further apart in euclidean space.
>
> **Ablation of the distance-weighted contrastive loss worsens silhouette score:** (Section 3.3) In contrast, all aspects of the model are crucial to generating clusters. For example, eliminating the distance-weighted contrastive loss leads to a 15% decrease in silhouette score for UCI Electricity, a 29% decrease for UCI Traffic, and a 53% decrease for Kaggle Retail. Although the online forecasting aspect of SpaRTen performed poorly in the Oxford stocks dataset, the clustering task beat the benchmarks. Eliminating the distance-weighted contrastive loss led 51% decrease in silhouette score on this task. We hypothesize that adding a spatial loss improves sub-network diversity by increasing state separation, aiding downstream clustering tasks.
>
> **Clinical Significance of Generated Clusters:** We have made significant changes to Figure 2 (see re-uploaded figure) to more clearly show the interpretable clustered states learned by SpaRTeN from clinical waveform data. We have also described interpretations of these states in Section 3.4 (Time is brain: Clinically Relevant Representation Learning).
>
> 2. What is the benefit of using a two-dimensional discrete state-space in the S-block? For the purpose of selecting the best base predictor, one-dimensional state-space should be enough. Also in the training of the S-block, cross-entropy loss function is used which actually works fine with the one-dimensional state-space.
>
> **Two-dimensional state space:** The benefit of the two-dimensional state-space is two-fold: first, ablation studies (Section 3.3) show it improves the clustering ability of the network, which we measure by silhouette score. Utilizing a one-dimensional state space is equivalent to eliminating the distance-weighted contrastive loss, and we show that this reduces the silhouette score of the network. Second, the two-dimensional state-space allows for states to exist multi-dimensionally. For example, in the context of intracranial pressure waveforms, there are two dimensions - directionality and stability. One dimension can capture the deterioration of intracranial pressure, whereas the other dimension can capture the stability and variability of the waveform. We highlight the clinical relevance of both directionality and stability in our explanation of the clinical significance of waveform states. More work, however, can be done on the interpretability of higher-dimensional spaces to see if increasing dimensions can improve upon clinical interpretation of state, especially given the growing complexity of ICU waveforms and calls to quantify this complexity in biologically meaningful ways [1].
> David M Maslove, Benjamin Tang, Manu Shankar-Hari, Patrick R Lawler, Derek C Angus, J Kenneth Baillie, Rebecca M Baron, Michael Bauer, Timothy G Buchman, Carolyn S Calfee, et al. Redefining critical illness. Nature Medicine, 28(6):1141–1148, 2022.
>
> (response will be continued in next comment)

---

> ### Author Response · Authors · 2022-11-19
> **Author response (2/3)**
>
> (continued from previous comment)
>
> 3. In Sec. 3.3, why does ensemble weight sharing help for time series forecasting? If we wanna a diverse ensemble which can cover a broader state-space, should we limit the use of ensemble weight sharing which may lead to the mode collapse (common challenge in GAN).
>
> **Ensemble Weight Sharing:** This is a valid point, and we thank the reviewer for highlighting this issue. We have clarified this as follows:
> (Section 2.4) While mode collapse is a known problem, we find that the sharing of weights across the first few layers leads to robust performance as seen with the paradigm of transfer learning. This is quite unlike the mode collapse seen in the training of generative adversarial networks. Without the sharing of weights across the first few layers, we find that learning requires significantly more samples because the overall structure of the time series must be learned for each unit in the R-block, in addition to learning of the relevant state. In contrast, with weight sharing, we find that learning of the overall structure of time series can be done jointly, and state separation can be learned by each sub-network.
>
> 4. The S-block actually reminds me of the attention mechanism which can also be used to select the optimal state/predictor. It would be interesting to see discussion on comparison of S-block and attention. For example, what are the advantages of S-block over attention and gating in the temporal fusion transformer (TFT).
> **Key Differences Between TFT and SpaRTeN:** In terms of practicality, first, we show that the S-block outperformed the temporal fusion transformer in online forecasting. Second, there are two key differences between the TFT and SparTEn: 1) generated output, and 2) task flexibility. While TFTs generate a representation in the latent space, SpaRTEn clearly identifies the representation in the space of the time series (Figure 2c,d). Generating a representation in the same space as the time series allows for improved explainability, and therefore, intervention. For example, if a patient has an ICP waveform that belongs to the state where there is cerebral ischemia, then clinicians can make an intervention relevant to cerebral ischemia. Current TFTs based methods generate representations in a latent space, and the relevant clinical state must be extracted from additional data. With SpaRTEn, we can directly obtain clinically relevant waveforms, and demonstrate this in our section titled “Time is brain” and have additionally attached it above. Second, TFTs have been implemented primarily constrained to forecasting. In contrast, SpaRTEn can take advantage of the diverse potential loss functions for the R-Block and improve individual sub-network performance. Third, as a versatile ensemble, SpaRTEn can include transformer-based models as networks in the R-Block.  Fourth and finally, we benchmark against TFTs forecasting ability and show that SpaRTEn outperforms. We have included this analysis in the paper (Section 2.5.3).
>
> 5. In practice, individual LSTM has limited forecasting power compared with state-of-the-art (SOTA) algorithms for time series forecasting. More analysis on how ensemble of LSTMs could achieve better prediction performance will help reader to understand the contribution and novelty of this work.
> **Ensemble Prediction Performance:** There are two novel aspects to this ensemble network - first, we utilize a contrastive loss to boost state separation, and second, we tie state to time i.e. we make states temporally dependent. We utilize the novel contrastive function to promote diversity in auto-regressive networks that allows for state separation. This is particularly relevant in the medical setting - state separation allows for subnetworks to represent clinically different states rather than coalescing to produce the optimal output prediction.
> Second, we make the state temporally dependent. SpaRTEn is an ensemble dedicated for online learning. Online learning can capture relationships such as stationary versus non-stationary time series that are not easily ascertained in time-agnostic state-encodings. For example, in a stationary time series, we notice that after a state has stabilized, the loss function for networks belonging to that state slowly decreases. However, as the patterns associated with the state start diverging from the expected pattern, which occurs during non-stationary time series, we see that the network loss gradually increases until it reaches a threshold. At the threshold, the state transition occurs and a new network is selected to represent that given state. These relationships cannot be deciphered in time-step agnostic state-encodings.
>
> (response will be continued in next comment)

---

> ### Author Response · Authors · 2022-11-19
> **Author response (3/3)**
>
> (continued from previous comment)
>
> 6. Evaluations are limited and only done on two datasets which makes it less convincing. More experiments and comparisons with SOTA methods are encouraged
> **Empirical Validation for Online Forecasting:** We include four other real-world applications and datasets in response to this comment.. We include UCI Electricity, UCI Traffic, Kaggle Retail and the stocks dataset. We show that in three out of the four datasets, SpaRTEn outperforms other convolutional-based methods such as N-Beats and attention based methods such as temporal fusion transformers and autoformers. We show that these methods additionally work for silhouette score and unsupervised clustering algorithms. We have moved these comparisons, which were originally in an appendix, to sections 3.1 and 3.2 of the main paper.
>
> (Sections 3.1 and 3.2)
> We benchmark against SOTA online forecasting models with convolutional approaches such as N-Beats [1] and attention based methods like Temporal Fusion Transformers [2] and Autoformers [3], where SpaRTEn outperforms on three out of the four datasets (Table 1).
> In order to compare the distinct properties of each state and demonstrate the ability to cluster waveforms within a given state, we train SpaRTeN with a latent state space of 3 x 3. We train a KNN with k = 9 on the 9 (3 x 3 = 9) distinct waveforms aggregated by state, and evaluate its ability to cluster all the waveforms on the dataset using a silhouette score, which is widely used to evaluate the goodness of a clustering technique. SpaRTEn outperforms all other methods used to cluster time series on all four datasets.
> 1. Oreshkin, Boris N., et al. "N-BEATS: Neural basis expansion analysis for interpretable time series forecasting." arXiv preprint arXiv:1905.10437 (2019).
> 2. Lim, Bryan, et al. "Temporal fusion transformers for interpretable multi-horizon time series forecasting." International Journal of Forecasting 37.4 (2021): 1748-1764.
> 3. Wu, Haixu, et al. "Autoformer: Decomposition transformers with auto-correlation for long-term series forecasting." Advances in Neural Information Processing Systems 34 (2021): 22419-22430.

---

### Official Review · Reviewer_9w1q · 2022-10-24

**Confidence:** 3
**Correctness:** 2
**Technical Novelty And Significance:** 2
**Empirical Novelty And Significance:** 2
**Recommendation:** 5

**Clarity, Quality, Novelty And Reproducibility:**

Please refer to the review section.


**Strength And Weaknesses:**

Please refer to the review section.


**Summary Of The Paper:**

Spatial Resolved Temporal Networks (SpaRTeN) are described in this paper as a novel, composite deep learning model for online, unsupervised representation learning in a spatially constrained latent space. In the proposed approach, two distinct components are simultaneously learned: a recurrent neural network ensemble that captures states in high-frequency time series, as well as a spatial block that is able to spatially resolve state changes based on the predictions generated by the other network. Throughout, it is an interesting paper, and the proposed method yields several practical applications for a wide range of machine learning problems. Major concerns and minor comments are presented in the review section.

**Summary Of The Review:**

a) The empirical studies do not provide convincing evidence. The proposed method can be applied to time series in a variety of applications, as indicated in the abstract. Consequently, it would be beneficial if authors benchmarked the proposed technique against several real-world datasets.

b) The performance of the proposed method should be compared with similar state-of-the-art approaches. That would be interesting to compare the proposed method with some recent studies in Transformers, Energy-Based approaches, and Reinforcement Learnings.

c) The figures and tables in this paper are ambiguous. As an example, what are the metrics in Table 1 (mean column) and Table 2? Furthermore, Figure 2 is blurry and difficult to read.

d) In this paper, the notations are confusing. In regular papers, scalers are denoted by small letters; vectors are defined with small letters (highlighted in bold), and matrices are denoted by capital letters using bold. In this paper, they are a lot of conflicts. It is so hard to trace what is a set, a matrix, or even a distribution.

f) There are some minor linguistic and typo problems in this paper.

---

> ### Author Response · Authors · 2022-11-19
> **Author response**
>
> We thank the reviewer for their constructive comments, and we hope we have provided sufficient responses. For this review, we have focused on a) including empirical validation across multiple datasets on standard time series datasets across state-of-the-art forecasting and clustering methods, and well as qualitative evidence for the clinical relevance of the generated states.
>
> a) The empirical studies do not provide convincing evidence. The proposed method can be applied to time series in a variety of applications, as indicated in the abstract. Consequently, it would be beneficial if authors benchmarked the proposed technique against several real-world datasets.
>
> **Empirical Validation for Online Forecasting:** We include four other real-world applications and datasets in response to this comment. We include UCI electricity, UCI traffic, Kaggle retail and the stocks dataset. We show that in three out of the four datasets, SpaRTEn outperforms other convolutional-based methods such as N-Beats and attention based methods such as temporal fusion transformers and autoformers. We have moved these from the appendix to Sections 3.1 and 3.2.
>
> **Qualitative Evidence for SpaRTeN:** We have made significant changes to Figure 2 (see re-uploaded figure) to more clearly show the interpretable states learned by SpaRTeN from clinical waveform data. We have also described interpretations of these states in Section 3.4 (Time is brain: Clinically Relevant Representation Learning).
>
> b) The performance of the proposed method should be compared with similar state-of-the-art approaches. That would be interesting to compare the proposed method with some recent studies in Transformers, Energy-Based approaches, and Reinforcement Learnings.
>
> **Model Comparisons:** As far as transformers are concerned, we compare to auto-formers and temporal fusion transformers, which are state of the art for interpretable online forecasting. These have been moved to sections 3.1 and 3.2 from the appendix. Energy-based approaches are generally more utilized for generative based model, which have a different task objective then the one we model. Reinforcement learning usually requires a reward function with a state and action, and there’s no action or reward parameterized in the way that reinforcement learning usually conceives it. We think that future work could extend energy-based approaches or re-parameterize the objective as a reward function with state action pairs, but that is beyond the scope of this paper.
>
> c) The figures and tables in this paper are ambiguous. As an example, what are the metrics in Table 1 (mean column) and Table 2?
> They’re in the title. We have moved them to the column names.
> Furthermore, Figure 2 is blurry and difficult to read.
>
> **Clarity:** Thank you for highlighting these issues, we have adjusted Tables 1 through 3 to clearly specify the names of metrics in the header. We have additionally made figure 2 larger and more visible for the viewer.
>
> d) In this paper, the notations are confusing. In regular papers, scalers are denoted by small letters; vectors are defined with small letters (highlighted in bold), and matrices are denoted by capital letters using bold. In this paper, they are a lot of conflicts. It is so hard to trace what is a set, a matrix, or even a distribution.
>
> **Notation:** We have changed all the scalars to small letters, all the vectors to small letters in bold. We have also provided clarifications after equations, as needed.
>
> f) There are some minor linguistic and typo problems in this paper.
>
> **Typos:** We apologize and have fixed grammatical and typographical errors that were present in the paper.

---

### Official Review · Reviewer_8297 · 2022-10-24

**Confidence:** 3
**Correctness:** 2
**Technical Novelty And Significance:** 2
**Empirical Novelty And Significance:** 2
**Recommendation:** 3

**Clarity, Quality, Novelty And Reproducibility:**

The overall paper description lacks clarity, which makes it really hard for me to understand the proposed method.
The authors should describe how their proposed approach is different from other contrastive learning techniques, such as BYOL or for biomedical applications such as CLOCS.
Moreover, proposed the proposed metrics for evaluation of the technique are very hard to interpret, and does not allow to judge the “clinical” usefulness of their approach.


**Strength And Weaknesses:**

Strengths:
1.	Use of different databases and applications (though does not see the clinical usefulness of the biomedical application, would have been interesting to assess on a classification problem (such as sepsis prediction for ICU data)
2.	The authors performed an ablation to highlight the added-value of each part of their technique.
3.	Source code is provided for better reproducibility.
Weaknesses:

1.	The authors could have given more insight on the practical usefulness of their framework, for instance with the interpretation of clinical states with the ICP waveforms. Is the method clinically relevant given the low silhouette scores obtained? Could be interesting to visualize ECG forecasting. What more does SpaRTEn bring compared to the other methods, as it is hard to tell with sMAPE values only. Are the predicted ECGs still interpretable (clear delineation of P, QRS, T waves …)? I don’ t understand what it means to predict 20 ECG samples (what is the sampling frequency? Given the typical ECG sampling frequency, 20 samples would correspond to less than 100ms, so I fail to understand the clinical usefulness of predicting such a short window of ECG samples
2.	The description of the proposed method lacks clarity. I would suggest present the methodology of the technique in a more structure way, clearly explaining how he training process is performed.
3.	I fail to understand the contrastive learning approach of the method. Do the authors suggest that the spatial representation is time independent? Could the authors compare their approach to a BYOL approach?  On top of that, one of the characteristics of biomedical times-series lies in their non-stationarity nature which would imply that the assumption of stable spatial representation does not hold. Could the author comment on that?


**Summary Of The Paper:**

The authors introduce a deep learning framework to analyze time series (identification of different states, detection of state transition, etc.). The framework presented in the paper is based on an ensemble network composed of LSTMs (each corresponding to a state) and a 1D CNN that allow to choose the best LSTM or state. The proposed technique has been applied for forecasting and clustering tasks on various time series data.

**Summary Of The Review:**

The presentation of the proposed method lacks clarity, which makes it hard to understand their approach, how it differs from state-of-the-art. The results do not allow me to judge the usefulness of the proposed technique.

---

> ### Author Response · Authors · 2022-11-19
> **Author response (1/2)**
>
> We would like to begin by thanking the reviewer for their insightful and constructive comments. On the whole, we have sought to a) highlight the clinical relevance of this work, and b) improve the clarity on the training process, and c) emphasize the difference between time-dependent and time-agnostic representation learning.
>
> 1. The authors could have given more insight on the practical usefulness of their framework, for instance with the interpretation of clinical states with the ICP waveforms. Is the method clinically relevant given the low silhouette scores obtained? Could be interesting to visualize ECG forecasting. What more does SpaRTEn bring compared to the other methods, as it is hard to tell with sMAPE values only. Are the predicted ECGs still interpretable (clear delineation of P, QRS, T waves …)? I don’ t understand what it means to predict 20 ECG samples (what is the sampling frequency? Given the typical ECG sampling frequency, 20 samples would correspond to less than 100ms, so I fail to understand the clinical usefulness of predicting such a short window of ECG samples.
>
> **Clinically Relevant:** Based on this comment, we have sought to narrow the focus of our paper to ICP waveforms because they are more clinically interpretable than ECG waveforms over the prediction window of our algorithm, and allow for an easier explanation. We include the clinical relevance of each of the states generated by SpaRTEn on intracranial pressure waveforms. Subsequently, we have made significant changes to Figure 3 (see re-uploaded figure), and we have added detailed description of clinical interpretation to the text (Section 4.4), of which we reproduce an excerpt here:
> In the second row, we start to see acute cerebro-vascular dysregulation. At point (1,0), we see that there are signs of instability, followed by a complete over-compensation, and an acute drop in the intracranial pressure. A patient in this state may warrant a CT scan to detect an early aneurysmal rupture. At point (1,1) and (1, 2), we see that there is complete dysregulation of the brain's vasculature with dramatic decreases and increases in intracranial pressure, respectively. These two waveforms are adjacent to each other and highlight intracranial pressure waveforms in a pathological state. In the context of waveform (1, 2), we might clinically witness a hemorrhage. A hemorrhage can increase local volume of blood, and decrease intracranial pressure. If a patient is in a hemorrhagic state such as an intracranial hemorrhage, interventions include endotracheal intubation to protect the airway, blood pressure management and hypertonic saline to reduce intracranial pressure
> [5]. In (1, 1), we see a rapid decrease in intracranial pressure as might be expected following treatment such as the placement of an external ventricular drain [6]. Notably, the waveform in (1,1) finishes  closer to the baseline state than that in (1,2), which makes sense because (1,1) involves a treatment designed to restore physiologic state.
>
> 1. Caceres, J. Alfredo, and Joshua N. Goldstein. "Intracranial hemorrhage." Emergency medicine clinics of North America30.3 (2012): 771-794.
> 2. Kramer, Andreas H. "Critical ICP in subarachnoid hemorrhage: how high and how long?." Neurocritical Care 34.3 (2021): 714-716.
>
> Comment 2. The description of the proposed method lacks clarity. I would suggest present the methodology of the technique in a more structure way, clearly explaining how he training process is performed.
>
> We thank the reviewer for highlighting the lack of clarity. We have made substantial changes to Figure 1 by color coding the steps for each of the blocks, and emphasizing the forward and back-propagation steps. Second, we have expanded on our description of the training process in section 2.3 (Loss Functions and Training with Backpropagation).
>
> (continued in next comment)

---

> ### Author Response · Authors · 2022-11-19
> **Author response (2/2)**
>
> (continued from previous comment)
>
> 3. I fail to understand the contrastive learning approach of the method. Do the authors suggest that the spatial representation is time independent? Could the authors compare their approach to a BYOL approach?  On top of that, one of the characteristics of biomedical times-series lies in their non-stationarity nature which would imply that the assumption of stable spatial representation does not hold. Could the author comment on that?
>
> **Stationarity vs Non-Stationarity:** This is an important question that needs to be clarified in the text. The key difference in our method is that our spatial representation is time-dependent rather than time-independent. In contrast to BYOL and other representation learning methods, this time-dependence allows us to build and generate state representations that are more representative of actual ICU states i.e. a patient will slowly deteriorate over time rather than abruptly change states each time step. We have added two paragraphs to Section 2.5.1 to address these key differences. Finally, we believe that extending the implementation of the BYOL method to online temporal tasks would be an interesting project in its own right, and would look forward to making this a future step in our research.
>
> (Excerpt from Section 2.5.1)
> First, SpaRTeN learns relationships between states in an online, sequential manner rather than offline and batch-dependent. As a result, SpaRTEn generates states that are time-dependent, whereas implementations of BYOL are time-independent. When the learned network is time-independent, we may see a different state assigned to each sequential time point or punctuated changes in state rather than a gradual evolution. However, clinically relevant changes in state are usually time-dependent, and as such, SpaRTen is expected to generate more interpretable predictions. For example, a patient who is developing an infection deteriorates over time until reaching a state of sepsis, rather than abruptly changing from a healthy to a septic state.
>
> Second, online learning captures relationships such as stationary versus non-stationary time series that are not easily ascertained in time-agnostic state encodings. For example, in a stationary time series, we notice that after a state has stabilized, the loss function for network belonging to that state slowly decreases. In non-stationary time series, patterns associated with the state start diverging from the expected pattern and the network loss gradually increases until it reaches a threshold. At the threshold, a state transition occurs and a new network is selected to represent that given state, implicitly allowing the detection of the state change. These relationships cannot be deciphered in time-step agnostic state-encodings.

---

### Decision · Program_Chairs · 2023-01-20

**Decision:**

Reject

**Justification For Why Not Higher Score:**

The initial version of the paper has several drawbacks: lack of clarity, insufficient experiments, less than convincing case study. The revision have addressed the second aspect, in my opinion, but it's impossible to determine if the other two were addressed without the reviewers reading the updated version.

It is unfortunately all-too-common for reviewers to not respond to author comments, especially when all of them opted for rejection based on the first read, and the only thing the authors can really do about it is attempt to provide an initial version of the paper that is in good enough shape to obtain at least borderline scores and thus warrant further discussion. The reviewers are a randomly selected sample of the target audience for the paper, so if neither of them indicated an interest in accepting the paper, then ICLR might not be the venue for it.

**Justification For Why Not Lower Score:**

N/A

**Metareview: Summary, Strengths And Weaknesses:**

The paper proposes a "composite deep learning model for online, unsupervised representation learning through a spatially constrained latent space", using two blocks, a block that encodes states and a spatial block that learns the state transitions. The method is tested on forecasting and clustering. The strengths of the paper, as identified by the reviewers, include the extensiveness of the datasets on which the tests were performed and the ablation studies -- the authors have augmented their experiments following the review period. The weakness of this paper was its clarity, various aspects of the paper: the notation, motivating the use of the contrastive loss, connections between the blocks and other mechanisms such as attention, the clinical importance of the application. The authors have given extensive responses to the issues raised concerning clarity, which would result in considerable modifications to the paper. Unfortunately, the reviewers have not read the responses, so it is impossible to determine whether the issues were addressed. Given that all 3 reviewers were initially unconvinced, and the extensiveness of the changes results in a paper that is very different from the initial submission (and thus warrants a new review cycle), I have no choice but to reject this paper.

**Summary Of Ac-Reviewer Meeting:**

N/A